# Complement Activation in the Treatment of B-Cell Malignancies

**DOI:** 10.3390/antib9040068

**Published:** 2020-12-01

**Authors:** Clive S. Zent, Jonathan J. Pinney, Charles C. Chu, Michael R. Elliott

**Affiliations:** 1Wilmot Cancer Institute and Department of Medicine, University of Rochester Medical Center, Rochester, NY 14642, USA; charles_chu@urmc.rochester.edu; 2Department of Microbiology, Immunology, and Cancer Biology, University of Virginia, Charlottesville, VA 22908, USA; jjp2xr@virginia.edu (J.J.P.); mre4n@virginia.edu (M.R.E.); 3Center for Cell Clearance, University of Virginia, Charlottesville, VA 22908, USA

**Keywords:** complement, cytotoxicity, phagocytosis, monoclonal antibody, B-cell lymphoma, chronic lymphocytic leukemia (CLL)

## Abstract

Unconjugated monoclonal antibodies (mAb) have revolutionized the treatment of B-cell malignancies. These targeted drugs can activate innate immune cytotoxicity for therapeutic benefit. mAb activation of the complement cascade results in complement-dependent cytotoxicity (CDC) and complement receptor-mediated antibody-dependent cellular phagocytosis (cADCP). Clinical and laboratory studies have showed that CDC is therapeutically important. In contrast, the biological role and clinical effects of cADCP are less well understood. This review summarizes the available data on the role of complement activation in the treatment of mature B-cell malignancies and proposes future research directions that could be useful in optimizing the efficacy of this important class of drugs.

## 1. Introduction

Unconjugated monoclonal antibodies (mAb) have an important clinical role in the management of mature B-cell malignancies [1,2,3,4,5]. The major mechanism of action of the mAbs targeting CD20, CD38, and CD52 surface antigens on B cells is activation of innate immune cytotoxicity. mAb-opsonized B cells can undergo Fc-induced cellular cytotoxicity by fixed macrophages via antibody-dependent cellular phagocytosis (ADCP) or by NK cells through antibody-dependent cellular cytotoxicity (ADCC), two of the major mechanisms of innate immune cytotoxicity [6,7,8,9,10,11,12,13,14,15,16,17]. A third major mechanism of innate immune cytotoxicity is mediated via activation of complement, thereby promoting complement-dependent cytotoxicity (CDC). In this proteolytic cascade [18,19], the lymphocytes are killed after downstream generation and binding of membrane attack complexes (MAC) which permeabilize the cell membrane. The details of these reactions can be found within this special issue, Figure 1 in the review by Golay and Taylor, and Figure 16 in the review by Taylor and Lindorfer. The B cells can also be killed via activation of ADCP by effector cells through complement receptors (CR) binding to C3-derived fragments covalently attached to the surface of target cells (cADCP) (Figure 1) [19,20,21,22,23,24].

We will review the clinical data on the role of complement activation by mAb in the treatment of mature B-cell lymphoid malignancies and our current understanding of the role of activation of complement in killing malignant B lymphocytes.

## 2. Complement-Activating Therapeutic mAb

The development of rituximab, the prototype unconjugated chimeric (mouse Fab_2_/human IgG1 Fc) anti-CD20 mAb, was the culmination of a multi-decade effort to utilize mAbs to treat malignancies of the immune system and autoimmune disease [25,26,27]. Use of rituximab for the treatment of mature B-cell lymphoid malignancies (FDA approval 1997) caused a paradigm shift in treatment of B-cell lymphomas [26]. Rituximab monotherapy was tolerable and achieved durable responses in the treatment of indolent B-cell lymphomas but was not curative. Combination of rituximab with standard chemotherapy regimens as chemoimmunotherapy (CIT) significantly improved treatment outcomes, including survival, in aggressive diffuse large B-cell lymphoma which is a potentially curable disease [1,2]. This success was followed by a plethora of concomitant and sequential mAb-containing treatment regimens, some of which have significantly improved treatment outcome and patient survival [3,4,5].

Next-generation anti-CD20 mAbs were developed to overcome the perceived limitations of rituximab. The fully human IgG1 wild-type Fc mAb ofatumumab (FDA approved 2009) was selected for improved CD20 binding properties (decreased off rate) and proximity of binding to the cell membrane, both of which increased complement activation [28,29]. In contrast, the development strategy for the humanized anti-CD20 mAb obinutuzumab (FDA approved 2013) was to optimize NK cell-mediated ADCC [30]. This was achieved through glycoengineering to defucosylate the human IgG1 Fc carbohydrate moiety, which substantially increased Fc receptor (FcR) affinity [30]. Obinutuzamab is not an efficient complement-activating mAb [30]. There is minimal published direct comparative data on the clinical efficacy of rituximab, ofatumumab and obinutuzumab as monotherapies, in CIT, or in combination with other targeted therapies.

Alemtuzumab (FDA approval 2001), a humanized rat anti-CD52 mAb utilizing wild-type human IgG1 Fc [31], is highly effective at killing circulating B and T lymphocytes by activation of both complement- [32] and cell-mediated cytotoxicity [13]. Alemtuzumab is an effective monotherapy for relapsed/refractory chronic lymphocytic leukemia/small lymphocytic lymphoma (CLL) patients [33,34]. Unfortunately clinical utility was limited by short durations of response and increased risk of opportunistic infections secondary to T cell depletion [35,36]. Alemtuzumab therapy of CLL has been largely superseded by targeted small molecule inhibitors but it remains an important treatment option for other rare B-cell malignancies such as B-cell prolymphocytic leukemia [37].

Daratumomab (multiple myeloma FDA approval 2015) is a fully human IgG1 mAb targeting CD38 that could have a role in treatment of other CD38-positive B-cell malignancies [38]. Daratumumab was selected because of its ability to activate complement [39]. In contrast, isatuximab, the second anti-CD38 mAb FDA approved for treatment of plasma cell diseases, does not appear to induce significant CDC in primary multiple myeloma cells [40].

mAbs are now an integral component of targeted therapy of B-cell malignancies based largely on very promising data from clinical trials. However, a better understanding of the mechanisms of action of these valuable drugs and the B-cell characteristics that determine sensitivity and resistance to mAb-mediated cytotoxicity is needed to continue to improve their efficacy. This knowledge should then allow for rational development of combination therapy with other drugs. These data will be critical for designing and testing regimens that can exploit orthogonal cytotoxic mechanisms so as to eliminate resistant subclones of malignant B cells and improve patient outcomes.

## 3. Mechanism of Action of mAb

Although mAb-containing treatment regimens are now standard of care for many diseases, their mechanisms of action and how pathological B cells evade their cytotoxic effects are not well defined. The mAb described above have limited or no direct cytotoxic effects and their therapeutic effects result primarily from activation of innate immune cytotoxicity [21,41,42,43,44,45,46].

Complement activation results in generation of MAC that can cause target cell lysis or necrosis. This has been demonstrated in vitro for mAb with wild-type IgG1 constant regions including rituximab, ofatumumab, alemtuzumab and daratumumab [21,28,39,45]. Although mAb-induced CDC can rapidly (within minutes) kill a large fraction of targeted B cells in vitro, the therapeutic significance of this mechanism in treatment of human lymphoid malignancies is not well defined [21]. There is circumstantial evidence that mAb-induced CDC is partially responsible for clearance of circulating B cells in patients with B-cell malignancies, but the role of complement activation and killing of malignant lymphocytes via generation of the MAC in lymphoid tissues is poorly understood.

Initial in vitro studies of rituximab-mediated cellular cytotoxicity demonstrated circulating human mononuclear cell-mediated ADCC of rituximab-opsonized B cells [47]. Subsequent studies showed that the primary mediators of this in vitro ADCC were NK cells [44]. A recently published study suggests that circulating NK cells have a low cytotoxic capacity and that NK cell ADCC is unlikely to be an important mechanism of mAb-induced clearance of circulating malignant B cells [14]. In contrast, ADCP mediated by fixed macrophages in the spleen and liver (Kupffer cells) has high cytotoxic capacity and plays a major role in rituximab-mediated clearance of circulating B cells [9,10,11,12,13,14,15,16]. ADCP activated by rituximab Fc binding to macrophage FcR is well described, but the role of activation of ADCP via the binding of complement C3b and its derivates to CR3 and CR expressed by macrophages is less well understood [14,16,22]. The relative role of CDC versus cADCP in the clinical activity of rituximab and other complement-activating anti-CD20 mAb is also not known.

The recognition that the major cytotoxic mechanism of anti-CD20 mAb is the innate immune system has profound consequences. Foremost among these is the likelihood that the finite cytotoxic capacity of the innate immune system limits the therapeutic effect of each dose of mAb and that subsequent “exhaustion” could have deleterious effects on therapy. We recently reported that hypophagia, the autoregulatory macrophage feedback mechanism that limits phagocytic capacity [16], could in part explain the observed rapid but finite clearance of CLL cells from the circulation by rituximab and ofatumumab [48,49,50,51]. Recovery of phagocytic activity in vitro as measured by quantitative ADCP assays takes approximately 24 h [13,16], a time scale similar to that observed in clinical trials using high-frequency low-dose rituximab therapy [48,51]. The cytotoxic capacity of activated complement via CDC and cADCP [23,24] during mAb therapy for B-cell malignancies is not well defined. A better understanding of the cytotoxic capacity of cADCP and the mechanistic overlap with Fc-FcR-mediated ADCP will be critical to developing therapeutic regimens that optimize mAb therapy. The goal of this optimization would be to develop therapeutic regimens that optimize patient outcome by maintaining a continuous rate of innate immune cytotoxicity without inducing “exhaustion”.

## 4. Complement Activation and CDC

The therapeutic effect of mAb activation of complement is determined by mAb pharmacokinetics and pharmacodynamics, availability of complement, target cell antigen expression and intrinsic target lymphocyte susceptibility to complement-mediated cytotoxicity.

### 4.1. Pharmacokinetics

Human Fc wild-type IgG mAbs are cleared from the circulation by the liver and spleen in a manner similar to the mechanism of clearance of endogenous IgG and in addition by ligation to target cells [52]. In patients with circulating malignant B lymphocytes, the rate of clearance of the first dose of anti-CD20 mAb is proportional to the amount of circulating CD20 available for ligation [50]. Although there is a correlation between low post-treatment serum mAb concentration and poor clinical response [53], there are no mechanistic data showing that these low serum mAb levels are responsible for treatment failure. In patients treated with current standard doses of anti-CD20 mAb, serum levels are consistently higher than required to saturate target binding in the circulation [14,44,50]. This suggests that the lower serum concentrations seen in patients with poorer prognosis are indicative of higher tumor burden rather than an inadequate therapeutic dose and that increasing the dose of mAb is unlikely to have any beneficial therapeutic effect. In contrast, there are limited published data on mAb concentrations achieved in lymphoid tissue and the ability of mAb to achieve complement-activating concentrations in malignant lymphoid tissue has not been reported. The possibility that lymphoid tissue mAb levels are not sufficient to activate complement requires further evaluation.

### 4.2. Pharmacodynamics

mAb currently used to treat B-cell malignancies vary widely in their ability to activate complement. mAb that bind ligand epitopes closer to the cell membrane (e.g., ofatumumab) are more likely to mediate membrane capture of activated complement components. Although ofatumumab has similar binding affinity for CD20 compared to rituximab, it has a slower off rate, resulting in more durable binding and binding stoichiometry that causes closer association of multiple mAb Fc regions which further increases its ability to activate complement [29,54,55,56,57].

mAb ability to activate complement is determined by target cell membrane ligand density and the ability to generate the hexameric Fc configuration that efficiently activates C1q [55,58]. For example, complement activation by rituximab in CLL, a disease characterized by low cell membrane CD20 density, is considerably less than in other B-cell lymphomas with higher CD20 density such as follicular cell lymphoma. The type I (complement-activating) [59] anti-CD20 mAbs rituximab and ofatumumab bind cell membrane CD20 dimers at a ratio of 2:1 to form mAb superstructures, which then concentrate on the cell membrane (forming lipid rafts) to efficiently activate complement as shown in Figure 2 [55,57]. In contrast, the type II (non-complement-activating) [59] anti-CD20 mAb obinutuzumab binds CD20 at lower ratios of 1:2, does not induce lipid rafts or formation of mAb superstructures and is not able to efficiently activate complement [56,57,60].

In summary, characteristics of mAb that contribute to efficient activation of complement are high antigen affinity and low off rate, binding to antigen epitopes close to the cell membrane, antigen binding stoichiometry that facilitates Fc association, and lipid rafts.

### 4.3. Complement Levels

Patients with some B-cell malignancies (e.g., CLL) can have defective complement function which could decrease mAb-induced cytotoxicity [61]. Serum complement levels have been shown to decrease significantly after administration of mAbs that are efficient complement activators and this could limit treatment efficacy with repeat dosing [20,50,62]. However, in most of these studies, the complement levels did not decrease below levels known to be sufficient for activation of maximum complement levels in vitro, and rebound of complement levels to baseline did not result in additional clearance of circulating CLL cells despite adequate mAb levels. These data suggest that complement deficiency was not a major limiting mAb effect.

Therapy with fresh frozen plasma (FFP) has been reported to increase activity of rituximab in CLL patients but this was not formally demonstrated to be mediated by increased complement availability [63]. FFP contains a large number of proteins and it could be useful to determine what component (e.g., immunoglobulin) could have modified the mAb therapeutic effect.

### 4.4. Complement Regulatory Proteins (CRP)

CLL cells have low expression of the CRP CD46 and high expression of CD55 and CD59 [64]. Inhibition of CD55 or CD59 increases in vitro CDC of malignant B cells [64] and there is in vitro data suggesting that for B cells with high levels of CD20, increased expression of CRP could decrease CDC [44]. Use of a targeted CD59 inhibitor in a mouse lymphoma model decreased lymphoma growth [65] but this approach has not yet evolved into an established clinical modality. Of note, in vitro daratumumab-induced CDC of primary multiple myeloma cells did not correlate with expression of CRP [39]. In conclusion, current data suggest that CRP expressed by target B cells is unlikely to determine the therapeutic efficacy of mAb in B-cell malignancies.

### 4.5. mAb Antigen

CDC requires higher levels of mAb ligation of cell membrane antigens than ADCP [14,50,64]. Consequently, malignant B lymphocytes that generally express lower levels of a specific mAb antigen, such as CD20 in CLL cells, are less sensitive to CDC [64]. In addition, rapid (within hours) loss of B-cell membrane CD20 after initiation of anti-CD20 mAb therapy has been well documented in vivo and can cause acquired resistance to in vitro CDC [50]. The primary mechanism of antigen loss is trogocytosis, the transfer of cell surface molecules (ligand-mAb immune complex) from a donor cell (opsonized B cell) to effector cells (e.g., macrophage, monocyte, and granulocyte) [66,67,68]. We have recently shown that ADCP of opsonized B cells by macrophages occurs as a rapid burst of engulfment (<1 h) followed by a period of exhaustion (hypophagia) with minimal further phagocytosis for the subsequent 24 h [16]. However, data from in vitro experiments show that trogocytosis is a slower and ongoing process [69] and clinical trials suggest that decreased in CD20 levels of circulating B cells continues for at least 24 h after the first infusion of ofatumumab to reach levels of ~3% of baseline [50]. These data suggest that trogocytosis of immune complexes (CD20-mAb) from circulating B cells could be an ongoing process that occurs independently from ADCP. Loss of mAb ligand following initiation of therapy can also occur because of endocytosis of mAb-ligand by target cells [70] but this process is likely to be slower and of lesser magnitude than trogocytosis [69]. Apparent loss of antigen could also occur because of selective mAb-mediated cytotoxicity of target B cells with higher antigen levels. However, careful evaluation of target circulating cell population numbers and distribution of membrane CD20 before and after initiation of ofatumumab therapy did not show selected survival of a pre-existing subpopulation of CLL cells with low levels of CD20 [50].

### 4.6. Intrinsic Cellular Resistance

The terminal event of activation of the classical complement pathway by mAb is generation of MACs that insert in the cell membrane and cause cell necrosis or lysis. In vitro mAbs that are highly efficient at activating complement such as alemtuzumab are cytotoxic to over 90% of CLL cells [71]. Cytotoxicity could be significantly increased by addition of ofatumumab to alemtuzumab but over 1% of CLL cells still remained viable, demonstrating that these treatments are unlikely to be curative. CDC of nucleated cells is known to require generation of multiple MACs per cells and cells can recover by shedding MACs [43] but most of the viable cells in this study had high levels of MAC equivalent to those seen in cells undergoing apoptosis [71]. To further investigate this finding, we assessed complement activation in viable cells by measurement of iC3b and C9 and showed that the majority of these cells had evidence of complement activation and MAC generation that would be expected to be lethal [50,71]. The Taylor laboratory subsequently showed that in these resistant cells, MAC generation induced the expected calcium flux but that this was not lethal [72]. The mechanism of resistance remains undetermined but these data suggest that a subpopulation of circulating B cells from patients with CLL could be intrinsically resistant to CDC.

## 5. Complement Activation and cADCP

As already noted, mAb ligation on the target cell surface can also lead to deposition of complement fragments that trigger ADCP upon binding to CRs on phagocytes. While cADCP can mediate rapid clearance of mAb-ligated tumor cells in vitro, the relative contribution of cADCP to tumor cell clearance in vivo is likely a very important but still poorly understood killing mechanism of mAbs.

### 5.1. cADCP Signaling

Activation of complement by mAb on target cells leads to the covalent binding of C3 activation fragments (C3b, which can decay to cell-bound iC3b, C3d) that effectively opsonize the target cell surface. These opsonins can be recognized by CR1, CR3, and CR4 on phagocytes to trigger a form of engulfment that is biochemically distinct from FcγR-mediated ADCP [73]. CR3 and CR4 recognition of iC3b is the principal means by which macrophages mediate cADCP [74]. To date, CR3, also known as α_M_β_2_ or CD11b, is the best-studied cADCP receptor. The signaling events that trigger phagocytosis downstream of CR3 are distinct from those of FcγR signaling. Activation of Rho family small GTPases (e.g., Rac, RhoA, and CDC42) is critical for driving the cytoskeletal reorganization that leads to target cell internalization [73]. Unlike FcγR ligation, the activation of CR3 during cADCP results in robust activation of RhoA but not Rac1 or Rac2 [75]. Active RhoA stimulates the nucleation and polymerization of actin at the phagocytic cup via activation of Rho kinase and mDia [76,77]. With regards to the role of Syk in cADCP, Kiefer et al. used macrophages from Syk-/- mice to establish definitively that Syk, a tyrosine kinase essential for FcγR ADCP, is dispensable for cADCP [78]. To date, the mechanisms and consequences of target degradation post-engulfment in FcγR ADCP and cADCP have not been investigated in-depth.

### 5.2. Crosstalk between FcγR- and cADCP

Part of understanding the relative contributions of both FcγR- and CR-activated ADCP pathways in therapeutic mAb-mediated cell clearance involves understanding the extent of crosstalk between these two pathways [79]. Early investigations into the connectedness of FcγR- and cADCP suggested the presence of C3 reduced the bound IgG levels that were needed to activate FcγR-ADCP; and found FcγR-mediated engulfment was lower when CR3 was blocked [80,81,82]. Jongstra-Bilen et al. showed that the engagement of IgG to FcγR resulted in increased CR3 lateral movement and enhanced iC3b binding to CR3 [83]. Additionally, Shushakova et al. showed that C5a-receptor signaling upregulated transcription of the activating FcγRI and FcγRIII was a pivotal component of FcγR-mediated signaling [84]. On the other hand, Huang et al. showed that the crosstalk between FcγR- and cADCP was more complex, and that their synergistic relationship was dependent upon the FcγR subtype that was co-stimulated with CRs [85]. Even the involvement of Syk in cADCP remains under debate based on evidence that cADCP can result in Syk phosphorylation and expression of dominant-negative Syk can disrupt cADCP [86]. These findings suggest that a complex synergy exists between FcγR- and cADCP and that co-stimulation of certain components within each pathway may be integral for full activation. The extent of their crosstalk and signaling connectivity, however, remain to be fully elucidated.

### 5.3. Importance of cADCP in mAb Therapy

While in vitro studies have been useful in establishing the ability of different mAbs to activate complement and trigger cADCP by macrophages, our understanding of the role of cADCP in vivo is much less developed. Using C1q-deficient mice, Di Gaetano et al. showed that activation of complement was essential for anti-CD20 mAbs to control tumor burden in a lymphoma model [87]. Intriguingly, this phenotype is similar to that seen in FcγR-deficient mice treated with CD20 and HER2 mAbs [12,42]. Additionally, van Spriel et al. revealed that knocking out CR3 substantially reduced the efficacy of the mAb TA99 in a syngeneic melanoma model compared to wild-type mice [88]. While these studies support the idea of cADCP as a major cytotoxic mechanism of mAbs in vivo, future studies that measure cADCP in vivo are needed to establish the role of complement in mAb-mediated cell clearance. Finally, there is a paucity of data on cADCP in humans, although there is evidence that anti-CD20 mAb therapies are less effective in CLL patients with complement deficiencies [62].

## 6. Conclusions

The development of mAbs targeting B lymphocytes has significantly improved therapy for patients with B-cell malignancies. Despite the impressive empiric clinical data supporting the use of these mAbs both as monotherapy and in combination regimens, there is limited data about how mAbs cause malignant lymphocyte cytotoxicity and how a subpopulation of these cells resist this cytotoxicity. Available data support complement activation as an important innate immune cytotoxic mechanism for rituximab, ofatumumab, daratumumab, and alemtuzumab. We propose that additional studies of the roles of CDC and cADCP in the clinical efficacy of these mAbs are required to provide the data needed to optimize the use of these drugs; these studies should allow for the design of new drugs to enhance activity and overcome resistance. These investigations could focus on defining the cytotoxic capacity of the innate immune system in order to develop clinical regimens that prevent immune “exhaustion”. It will be important to examine how nucleated cells can survive MAC-mediated calcium fluxes, and to determine the role of cADCP which has only limited overlap (and thus cross resistance) with CDC and ADCP. Current mAb-based therapeutic developments include engineered Fc with enhanced ability to form hexameric complement-activating structures [89], and bispecific mAb that decrease complement inhibitory proteins [90]. An improved understanding of the role of complement activation in the treatment of B-cell malignancies will both guide the development of these classes of drugs and provide the data required to develop novel drugs to improve complement-mediated cytotoxicity.

## Figures and Tables

**Figure 1 antibodies-09-00068-f001:**
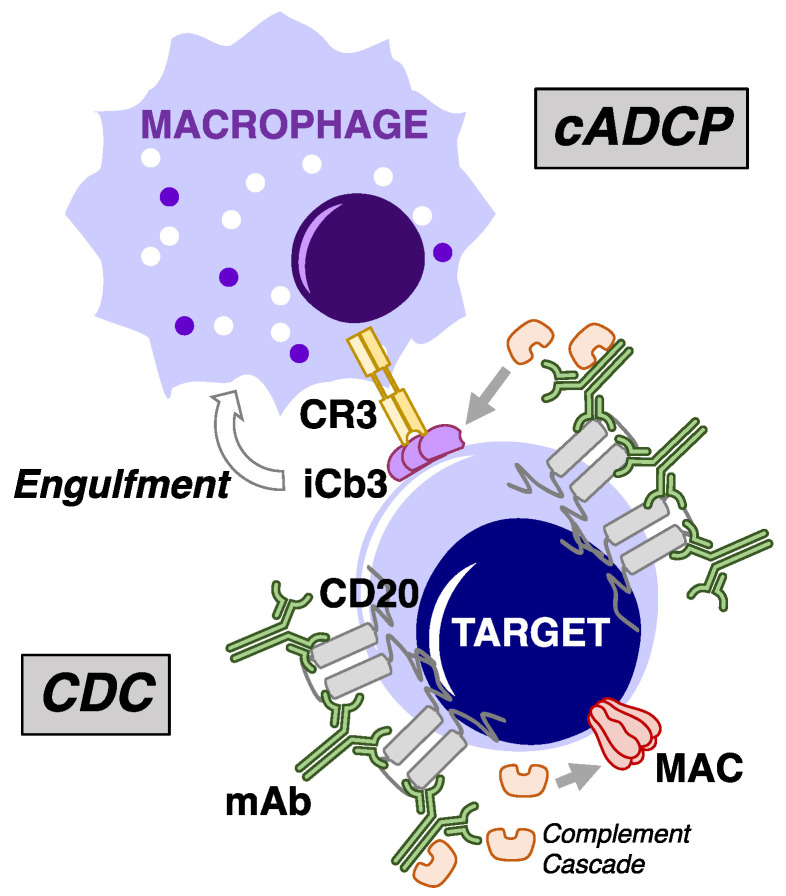
Overview of cytotoxic mechanisms underlying mAb-mediated complement fixation. Depiction of type I anti-CD20 mAb binding to surface of target cells. Complement-dependent cytotoxicity (CDC) occurs following formation and binding of multiple copies of the membrane attack complex (MAC) on the target cell surface downstream of mAb-induced initiation of the complement cascade. Target cell killing by complement receptor-mediated antibody-dependent cellular phagocytosis (cADCP) results from mAb-mediated deposition and covalent binding of C3 activation fragments to the cell surface, which are in turn recognized by complement receptors (CR3 is shown) which trigger activation of phagocytic pathways in phagocytes such as macrophages.

**Figure 2 antibodies-09-00068-f002:**
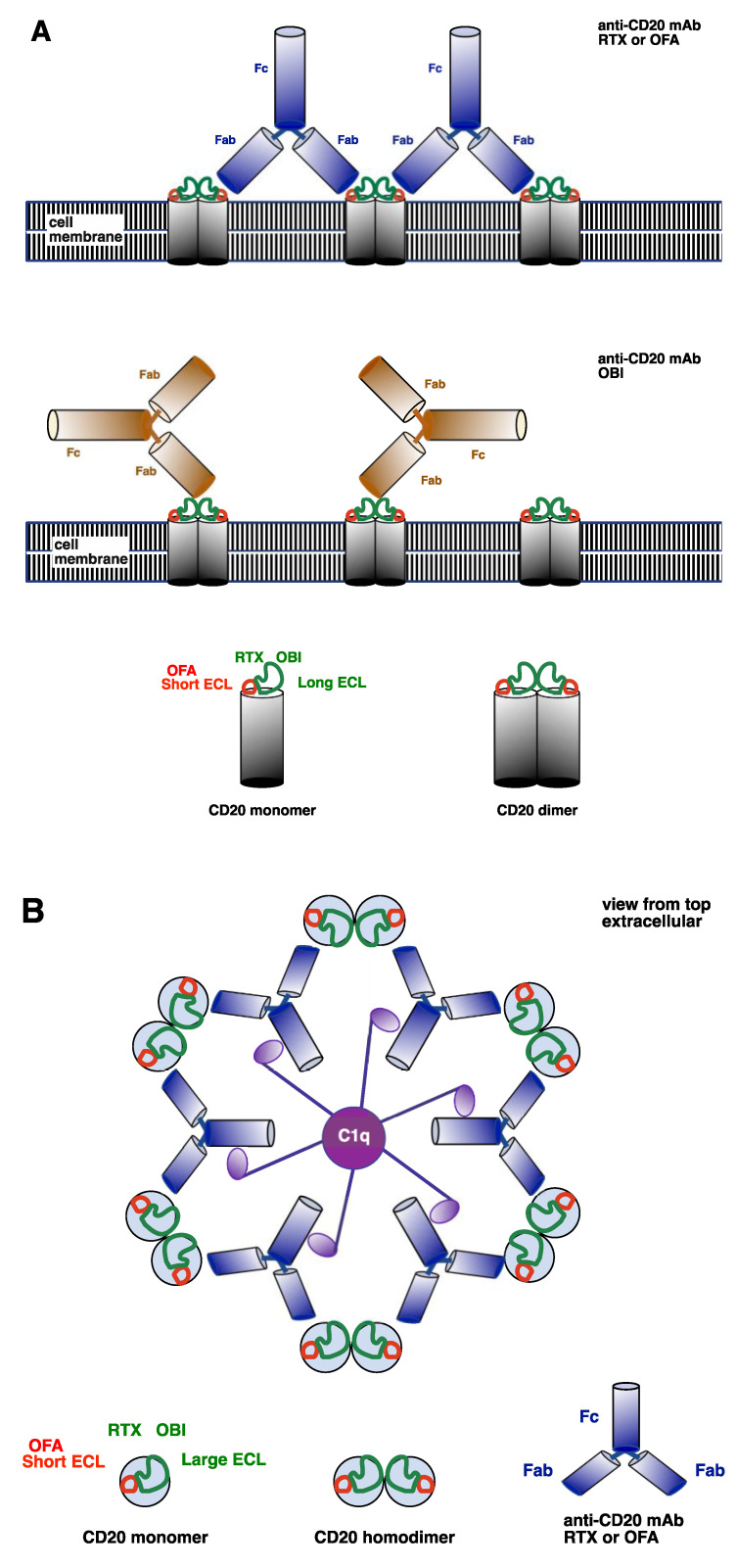
Anti-CD20 monoclonal antibody (mAb) ligation and activation of complement. (**A**) CD20 molecules form homodimers in the B-cell membrane. Rituximab (RTX) and obinutuzumab (OBI) ligate the long extracellular loop (ECL). RTX binds the long ECL in an area near the short ECL. OBI binds the long ECL in an area away from the short ECL. Ofatumumab (OFA) ligates the short ECL of the CD20 molecule. Type I complement-activating anti-CD20 mAb (RTX and OFA) Fabs can bind to two adjacent CD20 dimers. In contrast, the non-complement-activating anti-CD20 OBI binds only one CD20 dimer, resulting in a different Fc orientation to type I mAbs. (**B**) Model of the extracellular view from the top showing that type I anti-CD20 mAb RTX and OFA ligate adjacent CD20 dimers to form a hexamer that efficiently activates C1q.

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
