# Peer review of "Complement Activation in the Treatment of B-Cell Malignancies"

_2073-4468, 2020, doi:10.3390/antib9040068_

Round 1

Reviewer 1 Report

In this manuscript, the authors review the work and available data on several treatment strategies against B cell malignancies which involve the use of unconjugated monoclonal antibodies, leading to the activation of the complement system. This review is very well written and very informative. Figures serves the text well and are done very professionally. I believe this manuscript constitutes a timely summary of existing work on the topic. It fits very well the special issue on "The Role of Complement in Cancer Immunotherapy" and will certainly be of great interest to the readers of the Antibodies journal.

A serious concern I have with this manuscript is that it fails to mention another recent survey on a similar yet slightly larger topic:
Complement System: a Neglected Pathway in Immunotherapy. Bordron A, Bagacean C, Tempescul A, Berthou C, Bettacchioli E, Hillion S, Renaudineau Y. Clin Rev Allergy Immunol. 2020

Another important issue is that, for a general audience, it would be useful to expand the introduction with a paragraph about the mechanisms of complement activation. Not all readers will be familiar with the role of C3 fragments (such as iC3b) and complement receptors (such as C3). Without saying too much, the authors could just introduce the main mechanisms and refer the reader to one of the numerous reviews on the complement system (for example, one among those published by John Lambris).

Overall, the introduction feels quite short. I believe it should give some sense of perspective to the rest of the paper before we start digging into the data about mABs. It would be useful to construct the global picture within which mABs sit with respect to other treatments against B cell malignancies, along the lines of what it presented in the first and last paragraphs of the section on "Complement activating therapeutic mAb".

My other comments only relate to minor concerns. First, some sentences are quite complicated and should be reformulated (e.g., lines 70-72, lines 83-85, lines 184-187).

Second, there are several problematic typos:
- line 128: maintain -> maintaining
- line 133: medicated -> mediated
- line 152: bindin -> binding
- line 158: CD20 CD20 ???
- line 168: obinutuzumb -> obinutuzumab

Finally, the funding information should be reported in the funding section and not in the acknowledgements.

Author Response

The authors thank the reviewer for their critiques. Response and summary of changes to the manuscript are detailed below.

In this manuscript, the authors review the work and available data on several treatment strategies against B cell malignancies which involve the use of unconjugated monoclonal antibodies, leading to the activation of the complement system. This review is very well written and very informative. Figures serves the text well and are done very professionally. I believe this manuscript constitutes a timely summary of existing work on the topic. It fits very well the special issue on "The Role of Complement in Cancer Immunotherapy" and will certainly be of great interest to the readers of the Antibodies journal.

The authors thank the reviewer for this summary and appreciate the comments.

A serious concern I have with this manuscript is that it fails to mention another recent survey on a similar yet slightly larger topic: Complement System: a Neglected Pathway in Immunotherapy. Bordron A, Bagacean C, Tempescul A, Berthou C, Bettacchioli E, Hillion S, Renaudineau Y. Clin Rev Allergy Immunol. 2020

This reference has been added (ref. 19)

Another important issue is that, for a general audience, it would be useful to expand the introduction with a paragraph about the mechanisms of complement activation. Not all readers will be familiar with the role of C3 fragments (such as iC3b) and complement receptors (such as C3). Without saying too much, the authors could just introduce the main mechanisms and refer the reader to one of the numerous reviews on the complement system (for example, one among those published by John Lambris).

The introduction has been expanded with addition of references 18 (Lambris et al) and 19 and details of  two other articles in this special issue that provide a detailed description of complement activation.

Overall, the introduction feels quite short. I believe it should give some sense of perspective to the rest of the paper before we start digging into the data about mABs.

As detailed above, the introduction has been expanded.

It would be useful to construct the global picture within which mABs sit with respect to other treatments against B cell malignancies, along the lines of what it presented in the first and last paragraphs of the section on "Complement activating therapeutic mAb".

The authors appreciate the reviewers suggestion. However, this is a large topic which would require discussion of many mAb and B cell malignancies including those that do not activate complement and would need to be a separate review paper. This why we chose to detail the clinical benefits of the complement activating drugs in the body of the text. To ensure that readers can review some of the well documented benefits of mAb therapy, the appropriate references (1-5) have been added to the first sentence of the introduction (line 25).

My other comments only relate to minor concerns. First, some sentences are quite complicated and should be reformulated (e.g., lines 70-72, lines 83-85, lines 184-187).

Second, there are several problematic typos:

- line 128: maintain -> maintaining

- line 133: medicated -> mediated

- line 152: bindin -> binding

- line 158: CD20 CD20 ???

- line 168: obinutuzumb -> obinutuzumab

Finally, the funding information should be reported in the funding section and not in the acknowledgements.

These errors have all been corrected.

Reviewer 2 Report

Manuscript by Clive et. al. reviews an important topic in monoclonal antibody therapies. As much as it could of interest to the readers in the field, manuscripts has been complied without due rigor that it deserves. I ask for a through revisions for manuscript to be considered for publication. Below are some points.

  1. Make sentence shorter and clearer to readers especially the ones not experts in the field (I believe that is the purpose of writing a review: to de-mystify complex jargons). For line 37-40.
  2. Line 50 and everywhere else. Put ref before stop.
  3. There seems to be too much complexity defining and discussing ADCC, ADCP, CDC. Authors need to make discussion bit streamlined and discuss each of these in details. So need a thorough revising of manuscript. In current form, I find it confusing and difficult to follow.
  4. Line 133 – it should be mediated. Needs some proofreading as well. Same at 152. Revise 152-154. Writing is sloppy and must be improved. See line 158 as well.

Author Response

Manuscript by Clive et. al. reviews an important topic in monoclonal antibody therapies. As much as it could of interest to the readers in the field, manuscripts has been complied without due rigor that it deserves. I ask for a through revisions for manuscript to be considered for publication.

The authors consider this opinion unfortunate but are not able to share this reviewers concerns which are clearly at variance with reviewer 1.

Below are some points.

Make sentence shorter and clearer to readers especially the ones not experts in the field (I believe that is the purpose of writing a review: to de-mystify complex jargons). For line 37-40.There seems to be too much complexity defining and discussing ADCC, ADCP, CDC. Authors need to make discussion bit streamlined and discuss each of these in details. So need a thorough revising of manuscript. In current form, I find it confusing and difficult to follow

We have revised the manuscript to make it easier to read.

Line 133 – it should be mediated. Needs some proofreading as well. Same at 152. Revise152-154. Writing is sloppy and must be improved. See line 158 as well.

These errors have been corrected.

Round 2

Reviewer 2 Report

Concerns raised have been addressed by the reviewers